# Diversity and Pathogenicity of *Fusarium* Species Associated with Stalk and Crown Rot in Maize in Northern Italy

**DOI:** 10.3390/plants12223857

**Published:** 2023-11-15

**Authors:** Martina Sanna, Ilaria Martino, Vladimiro Guarnaccia, Monica Mezzalama

**Affiliations:** 1Department of Agricultural, Forest and Food Sciences, University of Torino, Largo Paolo Braccini 2, 10095 Grugliasco, Italy; martina.sanna@unito.it (M.S.); ilaria.martino@unito.it (I.M.); vladimiro.guarnaccia@unito.it (V.G.); 2AGROINNOVA—Interdepartmental Centre for the Innovation in the Agro-Environmental Sector, University of Torino, Largo Paolo Braccini 2, 10095 Grugliasco, Italy

**Keywords:** *Zea mays* L., *F. fujikuroi* SC, *F. nisikadoi* SC, *F. oxysporum* SC, multilocus sequence typing

## Abstract

The genus *Fusarium* includes several agronomically important and toxin-producing species that are distributed worldwide and can cause a wide range of diseases. Crown and stalk rot and grain infections are among the most severe symptoms that *Fusarium* spp. can cause in maize. Disease development usually occurs during germination, but it may also affect the later phases of plant growth. The purpose of this study was to investigate the diversity and pathogenicity of 41 isolates recovered from symptomatic seedlings collected in Northern Italy and seeds of five different geographical origins in 2019 and 2020. The pathogenicity was tested and confirmed in 23 isolates causing rotting in maize seedlings, with disease indexes from 20% to 90%. A multilocus phylogeny analysis based on four genomic loci (*tef1-α*, *rpb2*, *calm* and *tub2*) was performed on 23 representative isolates. Representative isolates were identified as species belonging to three species complexes (SC), including *Fusarium verticillioides* and *F. annulatum* in the *F. fujikuroi* SC. *Fusarium commune* was identified in the *F. nisikadoi* SC, and three different lineages were found in the *Fusarium oxysporum* SC. This study reports *F. annulatum* and two lineages of the *Fusarium oxysporum* SC as maize pathogens for the first time in Italy.

## 1. Introduction

Maize (*Zea mays* L.) is the first staple food in the world [1], and it represents the fifth-most produced commodity in the European Union (EU), supplying food, feed and fuel [2]. Italy represents the tenth maize producer in the EU, with 52,169,088 tons yielded in 2023 [3]. Italian production is concentrated in the Northern regions, thus representing an economically relevant sector of agriculture for that area. Several pathogens can affect maize, infecting seeds and seedlings and causing important plant diseases that lead to biosafety and phytosanitary problems and important yield and economic losses [4]. Stalk, crown and root rot are among the most severe diseases in maize [4]. Fungal species belonging to the *Fusarium* genus comprise one of the main causes of this disease in maize as well as in other cereals. *Fusarium* spp. are distributed worldwide and include a wide range of agronomically important and toxin-producing plant pathogens, which are causal agents of wilt, blight, tissues rot and cankers of many horticultural, ornamental and forest crops [5,6]. The infection occurs during seed germination and also affects the plant in later growth phases, causing severe diseases like root and stalk rot [7,8]. The disease can lead to premature senescence and lodging of the plants, with different levels of severity depending on the pathogenic species involved, the phenological stage of the plant and the environmental conditions that occur during the cropping cycle. *Fusarium* species are also able to produce a wide range of mycotoxins, which accumulate in the plant tissues during the infection process, posing an important risk to human and animal health [6,8,9].

In Europe, the main species involved with maize diseases are *F. graminearum*, *F. culmorum* and *F. proliferatum* [6]. Cases of root rot in maize are related to species of the *Fusarium fujikuroi* species complex (FFSC), especially *F. verticillioides* [10]. Species belonging to the *Fusarium oxysporum* species complex (FOSC) and the *Fusarium nisikadoi* species complex (FNSC) were frequently recorded in maize seeds and seedlings [11]. *Fusarium* mycelia can survive in maize residues and seeds, and they may colonize seedlings and plants through systemic infection [12]. Previous research reported the ability of *Fusarium* species to infect seeds, transmit the pathogen through the plant and become a source of infection of the roots and stalk up to the kernels [12,13,14]. The diagnosis of these diseases is often difficult due to the concurrent presence and multiple isolations of *Fusarium* pathogens from the same symptomatic portion of the plant [15].

Currently, more than 60 species belong to the FFSC; about 144 *formae specialis* are part of the FOSC; 6 species are included in the FNSC; and several species are not officially assigned to a species complex [16,17,18]. Difficulties in *Fusarium* spp. identification lie in their morphological features, which are usually strongly influenced by environmental conditions, and in their molecular profile because of wrong classifications of the sequences present in the public database and the nomenclature changes in the taxonomic system [19]. The molecular identification of fungi is usually obtained through sequencing of an internal transcribed spacer (ITS); however, in the case of the genus *Fusarium*, the ITS is exclusively able to discriminate the species complex, while the translation elongation factor (*tef1-α*) and the RNA polymerase second-largest subunit (*rpb2*) genomic regions are highly informative [20,21]. Also, the beta-tubulin (*tub2*) and calmodulin (*calm*) loci are used for *Fusarium* species identification [22]. Recently, the phylogenomic approach provided a high resolution to distinguish species within the *Fusarium* genus [19]. Thus, multilocus phylogenetic analyses combined with traditional identification based on morphological methods can deepen the knowledge of this genus.

The purposes of this work, considering the economic importance of maize and the impact of *Fusarium* species on this crop, are as follows: (i) to determine the pathogenicity of *Fusarium* spp. isolates obtained from maize seeds and seedlings, and (ii) to combine phylogenetic analysis with morphological characterization of the isolates to identify and understand the diversity of the *Fusarium* species affecting maize and causing stalk and crown rot in Northern Italy.

## 2. Results

### 2.1. Fungal Isolates

The observed symptoms in maize plants consisted of browning, wilting and collapse of the seedlings due to the decaying tissues of the stem. Disease incidence in the field was established considering the percentage of affected plants and ranged from 5 to 20% depending on the geographical location of the field. The symptoms were observed in seedlings of different maize hybrids already at the V1 stage. Rotting kernels covered by mycelia were observed in the incubation test. The recorded percentage of seeds infected with *Fusarium* spp. in the incubation test ranged between 5 and 56%. Forty-one isolates obtained from affected root, stem and crown tissue of the seedlings collected in the field and from the incubation test on seeds were identified as belonging to *Fusarium* spp. (Table 1).

### 2.2. Pathogenicity Test

A total of 36 out of 41 isolates tested caused root and crown rot like that observed in the field during spring 2019 and 2020 (Figure 1).

Different severity indexes, depending on the isolate tested, were observed. A total of 19 isolates showed disease indexes ranging from 13.3% to 46.7%, and only 17 of them showed a disease index higher than 50% (Table 2).

The identity of the reisolated fungi was proved by sequencing the *tef-1α* locus, confirming the Koch’s postulates. No symptoms were observed in healthy control plants. A total of 23 out of 36 pathogenic isolates were selected as representative isolates based on their cultural features, on which we proceeded with molecular analyses and characterization.

### 2.3. Phylogenetic Analyses

The preliminary analysis conducted on the obtained sequences showed that the 23 selected isolates belonged to three *Fusarium* species complexes, *Fusarium fujikuroi* SC, *Fusarium nisikadoi* SC and *Fusarium oxysporum* SC. The combined phylogeny analyses of *tef-1α*, *rpb2*, *calm* and *tub2* performed on FFSC isolates consisted of 101 sequences, including the outgroup sequence of *Fusarium foetens* (CBS 120665). A total of 2210 characters (*tef-1α*: 1–621, *rpb2*: 628–1185, *calm*: 1192–1726, *tub2*: 1733–2210) were included in the analysis; in the results, 563 characters were parsimony-informative, 604 were variable and parsimony-uninformative and 1025 were constant. A maximum number of 1000 equally most parsimonious trees were saved (tree length = 2973, CI = 0.602, RI = 0.812 and RC = 0.488). Bootstrap support values obtained with the parsimony analysis are shown on the Bayesian phylogenies in Figure 2. For the Bayesian analyses, the Dirichlet state frequency distributions were suggested by MrModeltest for analyzing all the partitions. The following models, recommended by MrModeltest, were used: GTR+G for *tef-1α*, SYM+I+G for *rpb2*, SYM+G for *calm* and HKY+G for *tub2*. In the Bayesian analysis, the *tef1-α* partition had 370 unique site patterns, the *rpb2* partition had 191 unique site patterns, the *calm* partition had 233 unique site patterns, the *tub2* partition had 269 unique site patterns and the analysis ran for 405,000 generations, resulting in 812 trees, of which 305 trees were used to calculate the posterior probabilities. In the combined analyses, eight isolates clustered with seven reference isolates of *F. verticillioides*, while six isolates were grouped with three isolates known as references of *F. annulatum* [19].

The combined phylogeny analysis of the three loci (*tef-1α*, *rpb2* and *calm*) performed on the FOSC isolates consisted of 47 sequences, including the outgroup sequence of *Fusarium udum* (NRRL22949). A total of 1762 characters (*tef-1α*: 1–589, *rpb2*: 596–1231, *calm*: 1238–1762) were included in the analysis; in the results, 77 characters were parsimony-informative, 171 were variable and parsimony-uninformative and 1502 were constant. A maximum number of 1000 equally most parsimonious trees were saved (tree length =  297, CI = 0.882, RI = 0.892 and RC = 0.787). Bootstrap support values obtained with the parsimony analysis are shown on the Bayesian phylogenies in Figure 3. For the Bayesian analyses, the Dirichlet state frequency distributions were suggested by MrModeltest for analyzing all the partitions. The following models, recommended by MrModeltest, were used: HKY for *tef-1α*, K80 for *rpb2* and *calm*. In the Bayesian analysis, the *tef1-α* partition had 109 unique site patterns, the *rpb2* partition had 71 unique site patterns, the *calm* partition had 57 unique site patterns and the analysis ran for 300,000 generations, resulting in 602 trees, of which 226 trees were used to calculate the posterior probabilities. In the combined analyses, one isolate clustered with four reference isolates and the ex-type of *F. nirenbergiae* and one isolate was identified as *F. cugenangense*, while five isolates were identified as *F. oxysporum sensu lato* because they did not cluster with any one of the reference sequences according to the recent taxonomy revision of this SC, reported by Lombard et al. [17].

The combined phylogeny analysis of the four loci (*tef-1α*, *rpb2*, *calm* and *tub2*) performed on the FNSC isolates consisted of 15 sequences, including the outgroup sequence of *Fusarium udum* (NRRL22949). A total of 2024 characters (*tef-1α*: 1–585, *rpb2*: 592–1362, *calm*: 1369–1594, *tub2*: 1601–2024) were included in the analysis; in the results, 186 characters were parsimony-informative, 333 were variable and parsimony-uninformative and 1487 were constant. A maximum number of 1000 equally most parsimonious trees were saved (tree length  = 616, CI  = 0.959, RI = 0.922 and RC = 0.884). Bootstrap support values obtained with the parsimony analysis are shown on the Bayesian phylogenies in Figure 4. For the Bayesian analyses, the Dirichlet state frequency distributions were suggested by MrModeltest for analyzing all the partitions. The following models, recommended by MrModeltest, were used: HKY for *tef-1α,* HKY+G for *rpb2,* JC for *calm* and SYM+G for *tub2*. In the Bayesian analysis, the *tef1-α* partition had 106 unique site patterns, the *rpb2* partition had 47 unique site patterns, the *calm* partition had 19 unique site patterns, the *tub2* partition had 57 unique site patterns and the analysis ran for 400,000 generations, resulting in 802 trees, of which 301 trees were used to calculate the posterior probabilities. In the combined analyses, two isolates clustered with seven reference isolates of *F. commune*.

### 2.4. Morphology

Morphological features, supported by phylogenetic analysis, were assessed and used to characterize six species belonging to three species complexes found in this study (Figure 5, Figure 6 and Figure 7).

Seven-day-old colonies of *F. verticillioides* showed white, abundant, aerial mycelia that developed violet pigments with age. The colony radius was 55–70 mm. Monophialides were produced and appeared in V-shaped pairs similar to “rabbit ears”. Microconidia were hyaline, oval- to club-shaped, aseptate, (6-)7-12(-13) × 2.5–3.5 μm (mean 7 × 3.0 μm), abundant in aerial mycelia and disposed in long chains. Macroconidia were straight and slender, with the apical cell foot-shaped, four to six septate, hyaline and (28-)32-49(-52) × 2.5–3 μm (mean 38.5 × 3.0 μm). Chlamydospores were absent.

*F. annulatum* colonies after 7 days at 25 °C on PDA reached 50–60 mm in diameter. The surface was characterized by white, aerial mycelia that became darker with age, while the reverse showed intense pink to purple pigments at the center of the colony. Conidiophores produced mono- and polyphialides, which generated a large number of microconidia that could be grouped in long chains on CLA. Microconidia were formed on aerial conidiophores, which were hyaline, oval to elliptical, aseptate and (2-)5-12(-15) × 1.5–3.5 μm (mean 8.8 × 2 μm). Macroconidia were hyaline, slender, straight to curved, with a foot-shaped apical cell and four to five septa and were (30-)35-42(-54) × 2–4 μm (mean 37 × 3 μm). Chlamydospores were absent.

The morphology of the *F. commune* colonies was characterized by white to pink, abundant, floccose to fluffy mycelia on the surface and violet pigmentation on the reverse colony. After 7 days of incubation at 25 °C, colony radial growth reach 45–50 mm on PDA. *F. commune* produced both mono- and polyphialides. On CLA, the isolates produced slightly curved, three to four septate macroconidia that were (23-)28-56(-66) × 2.5–6 μm (mean 38.5 × 4 μm) and aseptate, cylindrical and straight microconidia of (3.5-)5-7(8.2) × 2–3 μm (mean 6 × 2.5 μm). Chlamydospores were produced singly or in pairs.

*F. nirenbergiae* colony radial growth measured 55–60 mm after 7 days on PDA. The colony surface was characterized by abundant pink and floccose mycelia and by grayish-pink pigments on the reverse. Conidiophores carried on the aerial mycelia produced monophialides that bore oval, aseptate microconidia that were (8-)9-15(-16.2) × 2–3.5 μm (mean 11.2 × 3.2 μm) and three to four septate, slender, straight macroconidia with a papillate apical cell and a foot-shaped basal cell of (26.5-)28-30(-32.2) × 2.5–4.8 μm (mean 28.5 × 3.4 μm). Globose chlamydospores were produced.

One isolate was identified as *F. cugenangense*, and its colony morphology on PDA was characterized by white to pink, abundant, and cottony mycelia on the surface and pink-at-the-center to pale-gray on the colony’s reverse. The colony radius after 7 days at 25 °C under a 12 h photoperiod on PDA was 40–56 mm. It was characterized by monophialidic conidiogenous cells that produced three to six septate macroconidia that were (42.5-)46-55(-56.2) × 5.5–6.5 μm (mean 50.2 × 6 μm), with papillate apical cells and foot-shaped basal cells. Microconidia were abundant, oval to elliptical, zero to three septate and (7-)8.3-10.5(-13) × 4–7.5 μm (mean 9 × 5.6 μm). Chlamydospores were globose and formed singly or in pairs.

The isolates classified as *F. oxysporum sensu lato* were characterized by abundant pink to purple and floccose mycelia and purple to red pigments on the reverse. The colony radius was 50–60 mm after 7 days at 25 °C under a 12 h photoperiod on PDA. The isolates were characterized by conidiophores that produced monophialides that bore slender, straight, three to five septate macroconidia with foot-shaped basal cells and papillate apical cells, and they were (29-)30-37(-44) × 3–4.5 μm (mean 35 × 3.8 μm). Microconidia were abundant, oval, aseptate and (5.5-)6-11(-15) × 2–3 μm (mean 9.2 × 2.5 μm). Single chlamydospores were formed.

## 3. Discussion

Several species of *Fusarium* represent a severe problem for cereal cultivation and production worldwide, causing relevant yield and economic losses and posing a serious threat to human and animal health due to their ability to produce mycotoxins [4].

In the present study, *Fusarium* spp. were isolated from maize seedlings with symptoms of root and crown rot in Northern Italy and from rotted kernels collected in five different countries with the aim of investigating their diversity and pathogenicity. Isolates from seeds were included because of the ability of *Fusarium* species to be seedborne and seed-transmitted [12,23], causing stalk, crown and root rot that can be observed in the field under favorable soil moisture and temperature conditions. A polyphasic approach was used to study the fungal isolates obtained from the affected plants including the analysis of multiple characters, since the morphological features alone, which represent the traditional identification method used for *Fusarium* spp. identification, are not enough to discriminate among species [16]. The combination of multilocus sequence analyses, pathogenicity data and morphological characteristics represents the best way to characterize fungi at the species level. According to O’Donnell et al. [24], the ITS region is not able to distinguish *Fusarium* species boundaries and for this reason was not considered in this study. The *tef1-α*, *rpb2*, *tub2* and *calm* loci were used for *Fusarium* spp. identification according to the previous phylogenetic analysis of the genus reported in the literature [16,17,19]. Six different species were identified in association with the infection of the crowns, roots and seeds of maize: *F. verticillioides* and *F. annulatum* belonging to the FFSC, *F. commune* belonging to the FNSC and three different lineages in the FOSC. The FFSC contains 84 described species including a large number of cryptic species identifiable only based on phylogenetic inference [16,18,19]. The complex includes important plant pathogens and toxin producers [16], and species belonging to the FFSC can be discriminated from other complexes by their production of macroconidia, a large amount of microconidia and sporadic chlamydospores [19]. The results obtained in this study allowed for the classification of 14 isolates in this complex, identified as *F. verticillioides* and *F. annulatum*.

*Fusarium verticillioides* is one of the most important species that affects maize; it is distributed worldwide and can cause important yield and grain quality losses [25]. It is primarily reported as the causal agent of ear rot in maize; however, studies also reported the pathogen as responsible for symptoms of seedling decay and stalk, crown and root rot in maize [19,26,27]. *F. annulatum*, first described by Bugnicourt et al. [28], is a species associated with symptoms of rot in different crops, such as cantaloupe melons in Spain and saffron in China [29,30]. The name *F. annulatum* is often confused with *F. proliferatum*, a well-known maize pathogen associated with crown and root rot [15,31]. A phylogenetic analysis based on LSU, SSU and *tub2* genomic loci showed that the reference sequence of *F. annulatum* (CBS 258.54) introduced by Bugnicourt [28] clustered with representative strains of *F. proliferatum* (CBS 217.76, NRRL 25089) [32]. These results led to the wide report of *F. proliferatum* instead of *F. annulatum* as a maize pathogen. However, a recent multilocus phylogenetic analysis based on *calm*, *rpb1*, *rpb2* and *tef1-α* loci, including the epitype of *F. proliferatum* (CBS 480.96), established that this species clustered distantly from *F. annulatum* [19]. The same study demonstrated that several cereal pathogenic isolates, identified as *F. proliferatum* in previous research [15,31,33], should be identified as *F. annulatum*. The present research, based on the taxonomic characterization by Yilmaz et al. [16], demonstrated the characterization of the pathogenic isolates as the species *F. verticillioides* and *F. annulatum,* which belong to the same species complex and represent the highest proportion of the pathogenic isolates infecting maize samples considered in this study. To our knowledge, this is the first report of *F. annulatum* as a causal agent of stalk, crown and root rot in maize in Italy. *Fusarium commune* belongs to the FNSC, and it is principally known as a pathogen of rice and maize [34]. Its behavior as a pathogen is similar to that of some species belonging to the FOSC that cause rot and wilt in plants [19]. Recent studies reported *F. commune* as a causal agent of stalk, crown and root rot in maize in Italy [35] and in Liaoning Province in China [36]. The phylogenetic analysis conducted by Skovgaard et al. [37] identified the species as a sister group to the FOSC, a result supported by the high morphological similarity between these taxa. Species of the FNSC could be distinguished from those of the FOSC only because of the presence of long and thin monophialides and the occasional production of polyphialides [34,37]. To discriminate and identify the species, the *tef1-α* genomic region was used due to its high phylogenetic signal [34]. *Fusarium oxysporum* is an economically important soilborne and ubiquitous plant pathogen that occupies the fifth place in the top-ten ranking of the most important phytopathogens [38], and it is mainly known as a causal agent of plant wilts. The challenge in the identification of the species belonging to this complex is due to the inability to discriminate them on the basis of morphological features, the affected wide host range and their geographical distribution [39,40]. The *tef1-α* and *rpb2* genomic loci provided the best resolution in distinguishing the species, as seen by Lombard et al. [17]. The calmodulin locus provided a little support, while the beta-tubulin locus was excluded. Considering the current literature [17,19], the multilocus phylogenetic analysis performed in this study allowed us to identify seven isolates within three lineages of the FOSC. The first lineage includes one isolate that formed a well-supported clade with the reference isolate and the ex-type of *F. nirenbergiae*. The second lineage includes one isolate that clustered with the reference of *F. cugenangense*. The third lineage includes five isolates that did not cluster with any of the reference species used for the phylogenetic analyses and that were defined as *F. oxysporum sensu lato*. *Fusarium nirenbergiae* belongs to the FOSC, and it is reported as a pathogen on saffron in China [30] and on passionfruit in Italy [41]. It was recently described as a pathogen in maize in China [19], and our study represents the first finding of this species as a maize pathogen in Italy. It is closely related to *F. curvatum*, and it can be morphologically distinguished from this species by the production of monophialidic conidiogenous cells and the production of chlamydospores, which are absent in *F. curvatum* [17]. For species identification, morphological features must be supported by phylogenetic inference. The *tef-1α* and *rpb2* gene regions provided the best resolution to distinguish the species [17]. *Fusarium cugenangense* was previously included in the species *F. oxysporum* f. sp. *cubense*, the causal agent of banana wilt; however, phylogenetic analyses distinguished this lineage as a new, independent species [42]. This pathogen has a wide host range, such as *Acer palmatum*, *Crocus* sp., *Gossypium barbadense*, *Hordeum vulgare*, *Solanum tuberosum*, *Smilax* sp., *Tulipa gesneriana*, *Musa nana*, *Musa* sp., *Vicia faba* and *Zea mays* [18,19,42]. To our knowledge, this is the first report of *F. cugenangense* as a pathogen of *Zea mays* in Italy. This species is closely related to *F. callisthephi*, *F. elaeidis* and other *formae speciales*; however, it can be discriminated from the other species under the morphological point of view by the septation of the macroconidia and because it is the only one that produces monophialides [17,42]. Molecular identification and discrimination were supported by the amplification of the *tef-1α* and *rpb2* loci [17]. The identification of species belonging to the FOSC represents a great challenge due to the complexity and endless evolution of the taxonomy of the genus *Fusarium*. During the last decades, a plethora of new species were described, which increased the problems for *Fusarium* taxonomy users [43]. Therefore, there is an agreement on the need to stabilize the taxonomy of the complex while conducting further studies to clarify species concepts to allow the correct characterization of species within the FOSC [17,43,44]. The high species diversity, found in the present study from a molecular point of view, should be supported by analyses on the pathogenicity and host preference of these species.

The pathogenicity tests performed herein confirmed that all the species were able to cause symptoms of crown and root rot in maize seedlings. This is in line with the results obtained by other scientists that contribute to increasing the knowledge of the complexity of the maize microbiome and on the etiology of soilborne diseases [45,46,47]. The isolates that were confirmed as pathogenic showed different levels of aggressiveness in maize seedlings. The *F. verticillioides*, *F. annulatum* and *F. commune* isolates always showed a disease index higher than 50% except for one isolate of *F. verticillioides* (8.2), which showed a disease index of 20%. Regarding the isolates belonging to the FOSC, one isolate of *F. oxysporum sensu lato* and the isolate of *F. nirenbergiae* showed a disease index higher than 50%, while the other isolates of *F. oxysporum sensu lato* and the isolate of *F. cugenangense* showed lower indexes ranging from 20% to 45%. Considering the economic and agronomic relevance of maize and the susceptibility of this crop to pathogenic *Fusarium* species, it is important to provide a correct diagnosis for rapid and effective disease management. No specific antifungal products are available to control these pathogens in maize plants, but several studies investigated the efficacy of different chemical and biological products against *Fusarium* pathogens. Shin et al. [48] tested the efficacy of six chemical fungicides, showing the efficacy in vitro of tebuconazole, while other studies evaluated the antagonistic efficacy of two species of *Trichoderma* and a *Bacillus* strain against *Fusarium* species associated with stalk rot in maize [49,50]. This study investigated the species involved in maize diseases associated with symptoms of stalk, crown and root rot in Northern Italy as well as those associated with seeds from different countries. Moreover, it provides useful information on tools to analyze the target loci to identify *Fusarium* species, laying the base for future studies in their detection to develop specific and sensitive diagnostic tools that speed up the diagnosis of these pathogens. The identification process usually requires a long time and several steps, starting with the description of the symptoms, the environmental conditions in which the infection occurred and the isolation, purification and morphological and molecular identification of the causal agents of the disease observed [51]. The development of rapid, specific and accurate molecular diagnostic tools could allow for the identification and quantification of multiple pathogens in symptomatic plants and seeds as well as in those not yet expressing symptoms. Further investigations should be undertaken to evaluate the putative cross-pathogenicity of these species and the seedborne rate in causing the symptoms observed in the field and reproduced in this study to provide a deeper insight into the pathogens and disease development and then to improve the management of sustainable control strategies.

## 4. Materials and Methods

### 4.1. Fungal Isolates

During 2019 and 2020, different surveys were conducted in six maize fields in Northern Italy. The surveyed fields were in San Zenone degli Ezzelini (VI) (45°47′ N, 11°50′ E), Livorno Ferraris (VC) (45°16’ N, 8°5’ E), Cigliano (VC) (45°18′ N, 8°01′ E) and Crescentino (VC) (45°11’ N, 8°6’ E). Root and crown rot symptoms were detected in seedlings of different hybrids of maize early in the season between the V1 (first leaf) and V3 (third leaf) phenological stages. Symptomatic samples were collected and washed under running tap water for 2 min to remove soil debris. Small sections (0.1–0.2 cm) were cut on the edge of the symptomatic portions, surface-sterilized in 1% hypochlorite solution for one min, rinsed in sterile distilled water and placed on potato dextrose agar (PDA, Merck, Darmstadt, Germany) to isolate fungi. After an incubation of 72 h at room temperature, the plates were observed and mycelial plugs from the developed fungal colonies were transferred onto new PDA plates to obtain pure cultures.

In 2019, 24 commercial lots of different maize hybrids, certified following the OECD seed scheme by Crea-DC-I and produced in 5 different countries (France, Italy, Romania, Turkey and the USA), were provided by CAPAC (Soc. Coop. Agricola, Torino, Italy). A total of 500 g of seeds per lot was sampled and analyzed with an incubation test to evaluate their phytosanitary conditions [52]. A total of 400 seeds of each lot were disinfected with 100 mL of a water solution containing 55.9% of commercial chlorine (5.37%), 10.4% of absolute alcohol (96%) and 10 µL of Tween 20 for 15 min and then rinsed three times with sterile distilled water and dried on sterile paper. The disinfected seeds were placed in 12 × 12 plastic boxes over three layers of sterile filter paper soaked with a 0.05% sodium hypochlorite water solution. The boxes were placed in a growth chamber for 48 h at 25 °C ± 2 °C under a 12 h near-ultraviolet-light (NUV)/12 h dark cycle, then for 24 h at −20 °C and then incubated in the growth chamber for 11 days. Colonies were isolated from seeds and placed on PDA plates to obtain pure cultures.

Among the colonies obtained from the plant and seed material collected, only isolates morphologically similar to *Fusarium* spp. were used for the following analysis.

### 4.2. Pathogenicity Test

The pathogenicity of the 41 isolates was assessed following the protocol described by Okello et al. [15]. Pure cultures of the isolates were grown on PDA amended with 25 mg/L of streptomycin sulphate for 14 days at room temperature. After two weeks, mycelium plugs (15 mm) of each isolate were transferred into conical flasks (250 mL) containing a sterile sand/cornmeal substrate, which was prepared with 54 g of sand, 6 g of cornmeal and 10 mL of deionized water per flask. Five replicate flasks were used for each isolate. The inoculated flasks were then incubated at 23 ± 2 °C for 23 days, and they were mixed daily. A total of 300 maize seeds (P1565, Pioneer Hi-Bred, Gadesco-pieve Delmona (CR), Italy) were incubated at 23 ± 2 °C for three days in Petri dishes filled with moisturized sterile filter paper to promote their germination and to obtain seedlings for inoculation. Once germinated, six seedlings per isolate were transplanted in inoculated pots (volume: 2 L) filled following the protocol described by Bilgi et al. [53] with a first layer of 40 g of perlite followed by a second layer of 20 g of inoculum and a final layer of 20 g of perlite. A total of 123 inoculated plastic pots were used, considering 2 seedlings per pot and 3 pots per fungal isolate. The pots were incubated in the greenhouse at 22 ± 2 °C for 14 days. The root rot severity was assessed with a scale that ranged from 1 to 5 at 14 days postinoculation. The adopted scale was as follows: 1 = germinated seed and healthy seedling without symptoms of root rot; 2 = germinated seed and 1–19% symptomatic roots; 3 = germinated seed and 20–74% symptomatic roots; 4 = germinated seed and >75% symptomatic roots; 5 = complete colonization of the seed and undeveloped seedling [54]. The data were expressed as disease index (DI) 0–100, calculated with the following formula: DI = [∑(i × ni)]/(4 × total of plants)] × 100, where i = 0–4 and ni is the number of plants with rating i. The assay was performed in triplicate and the data obtained expressed as mean value of the three replications carried out.

### 4.3. Statistical Analyses

The data were subjected to analysis of variance (ANOVA) after testing that the resulting disease index data were normally distributed with a Levene’s test using SPSS Statistics v. 27.0 (IBM Corp., Armonk, NY, USA). The Duncan’s test was used to explore differences between multiple group means (*p* ≤ 0.05). Statistical analysis was performed with the Statistical Package for Social Science (SPSS, IBM, Chicago, IL, USA) version 27.0.

### 4.4. DNA Extraction, PCR and Sequencing

A total of 23 isolates were selected as representative based on their positive results in the pathogenicity test and used for the following analyses. Genomic DNA was extracted from each isolate, transferring 100 mg of mycelium in a 2 mL microcentrifuge tube and following the manufacturers’ instructions of the Omega E.Z.N.A.^®^ Fungal DNA mini kit (Omega Bio-Tek, Norcross, GA, USA) after a 15 min cycle at 25 Hz in Tissuelyser (Qiagen^®^, Hilden, Germany). Partial translation elongation factor-1α (*tef-1α*), RNA polymerase second-largest subunit (*rpb2*), calmodulin (*calm*) and beta-tubulin (*tub2*) genomic regions were amplified using EF1 and EF2 [55], rpb2-7cr and rpb2-5f [56], CAL-228f and CAL-737r [57], CL1 and CL2A [24] and T1 [58] and Bt2b [59] primers, respectively. The PCR mixtures and the cycling conditions for the amplification of *tef-1α*, *calm* and *tub2* followed the protocols described by Guarnaccia et al. [60] and Weir et al. [61]. For the *rpb2*, the PCR protocol by Yilmaz et al. [16] was optimized as follows: 94 °C for 90 s; 40 cycles of 94 °C for 30 s, 55 °C for 90 s, 68 °C for 2 min; 68 °C for 5 min. PCR amplification was checked by electrophoresis on 1% agarose (VWR Life Science AMRESCO^®^ biochemicals, Milano, Italy) gels stained with GelRed^TM^. PCR products were sequenced by BMR Genomics (Padova, Italy), and the obtained sequences were analyzed and assembled with the program Geneious v. 11.1.5 (Auckland, New Zealand).

### 4.5. Phylogenetic Analyses

The sequences generated in this research were analyzed with the NCBI’s GenBank database through the BLAST-N program to determine the closest species and the species complexes to which they belong and then compared with reference sequences reported in the literature [16,17,18,19,24,39,44,62,63,64,65,66,67,68,69] and downloaded from GenBank to establish the identity of the explored isolates. All the different regions of the sequences in this study and those downloaded from GenBank were aligned with the MAFFT v. 7 online server (http://mafft.cbrc.jp/alignment/server/index.html, accessed on 4 September 2023) [70] and then manually adjusted in MEGA v. 7 [71]. A preliminary analysis was conducted on the tef1-α region to determine which species complex the representative isolates belonged to. Phylogeny was processed through different analyses conducted as multilocus sequence analyses using different datasets in accordance with previous studies [16,17,19]. The analyses for the FFSC and the FNSC were performed by combining *tef-1α*, *rpb2*, *calm* and *tub2* datasets, rooted with *F. foetens* (CBS 120665) and *F. udum* (NRRL 22949), respectively. The combined *tef-1α*, *rpb2* and *calm* datasets were used to perform the analyses for the FOSC, rooted with *F. udum* (NRRL 22949). The phylogenies were based on maximum-parsimony (MP) and Bayesian inference (BI) methods. The MP analyses were performed with PAUP [72], while the Bayesian analyses were carried out with MrBayes v. 3.2.5 [73], including the best evolutionary model for each partition as defined by MrModelTest v. 2.3 [74]. BI analyses were processed using four Markov chain Monte Carlo (MCMC) chains with a sampling frequency of 1000 generations. The heating condition was set to 0.2 and the analyses ended when the standard deviation of split frequencies was less than 0.01. For the MP analyses, phylogenetic relationships were estimated by heuristic searches with 100 random addition sequences. Tree bisection–reconnection was used with the branch-swapping option set to ‘best trees’ only, with all characters weighted equally and alignment gaps treated as fifth-state. Tree length (TL), consistency index (CI), retention index (RI) and rescaled consistence index (RC) were calculated for parsimony, and the bootstrap analyses were based on 1000 replications. The clade was supported when the bootstrap support value was ≥70%, and the Bayesian PP value was ≥0.9. Sequences generated and used in this study were deposited in GenBank (Appendix A).

### 4.6. Morphology

The characterization and description of *Fusarium* isolates was conducted using macro- and micromorphological features as described by Leslie et al. [25]. Single conidia colonies of the 23 representative isolates were grown on PDA for 10 days. Colony growth and macromorphological features were determined by placing agar plugs (5 mm) taken from the edge of actively growing cultures on PDA plates and incubating at 25 ± 1 °C under 12/12 h near-UV light for 7 days [69]. All the isolates were transferred onto carnation leaf agar (CLA) plates [75] and incubated at 25 ± 1 °C under 12/12 h near-UV light for 14 days to induce sporulation. Micromorphological features were observed, and 50 random measurements of macroconidia, microconidia, conidiogenous cells and chlamydospores were taken for each isolate at 40× magnification with a Leica DM2500 microscope. The observations were made by placing the plates directly under the microscope. Measurements were reported as mean value, standard deviation and minimum and maximum values.

## Figures and Tables

**Figure 1 plants-12-03857-f001:**
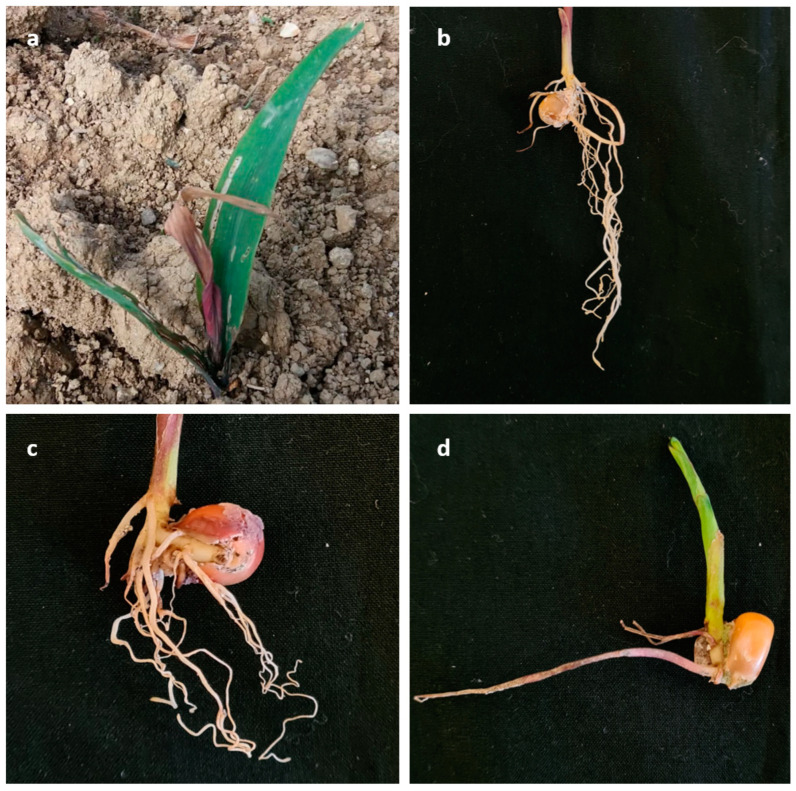
Symptoms caused by *Fusarium* spp. (**a**,**b**) observed in the field and (**c**,**d**) after pathogenicity trials on leaves, roots and crowns of maize seedlings.

**Figure 2 plants-12-03857-f002:**
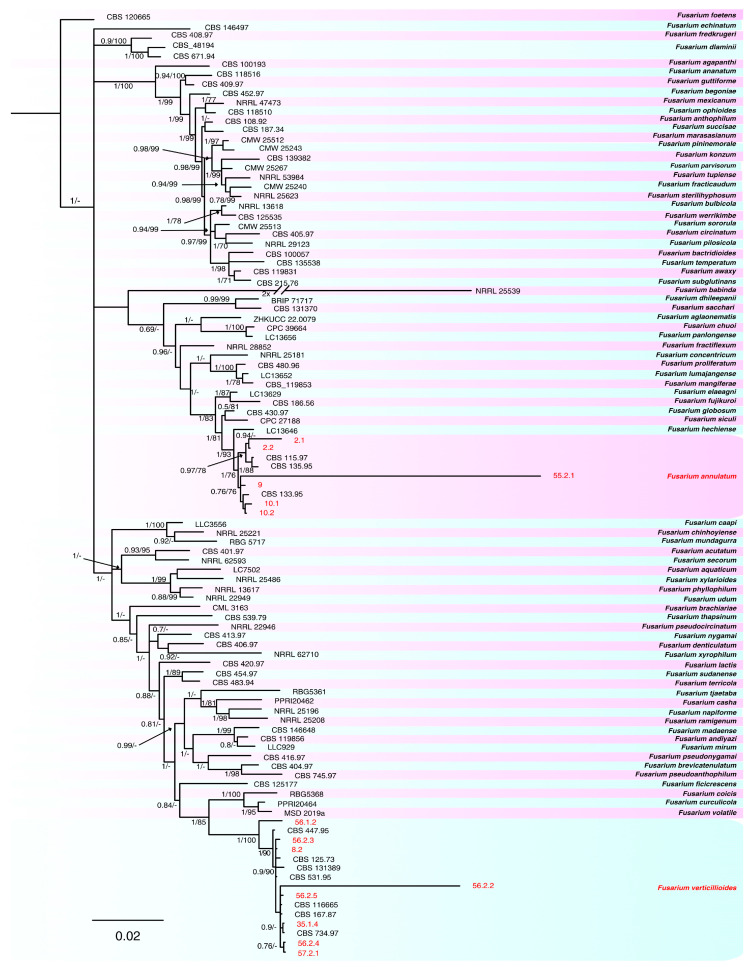
Consensus phylogram of 305 trees resulting from a Bayesian analysis of the combined *tef1-α*, *rpb2*, *calm* and *tub2* sequences of *Fusarium* spp. belonging to FFSC. Bayesian posterior probability values and bootstrap support values are indicated at the nodes. The isolates collected and species found in this study are indicated in red. The tree was rooted to *Fusarium foetens* (CBS 120665).

**Figure 3 plants-12-03857-f003:**
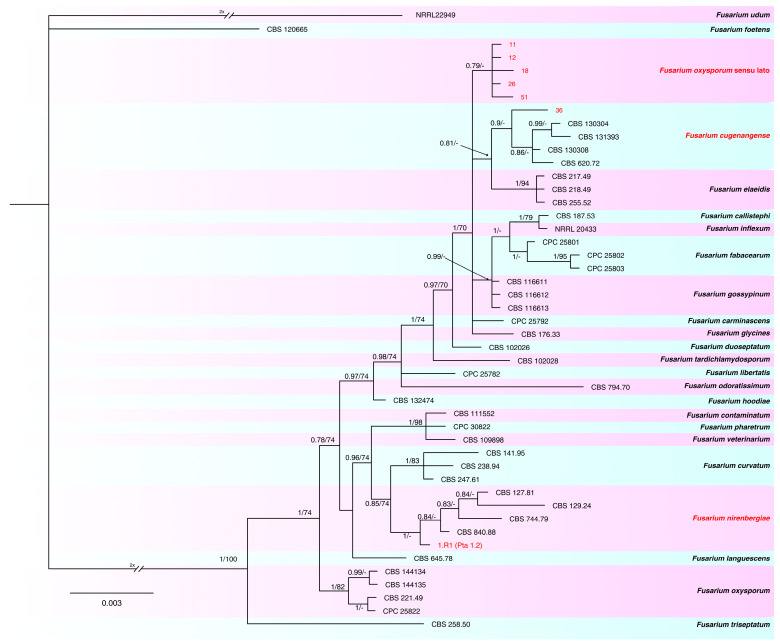
Consensus phylogram of 226 trees resulting from a Bayesian analysis of the combined *tef1-α*, *rpb2* and *calm* sequences of *Fusarium* spp. belonging to FOSC. Bayesian posterior probability values and bootstrap support values are indicated at the nodes. The isolates collected and species found in this study are indicated in red. The tree was rooted to *Fusarium udum* (NRRL22949).

**Figure 4 plants-12-03857-f004:**
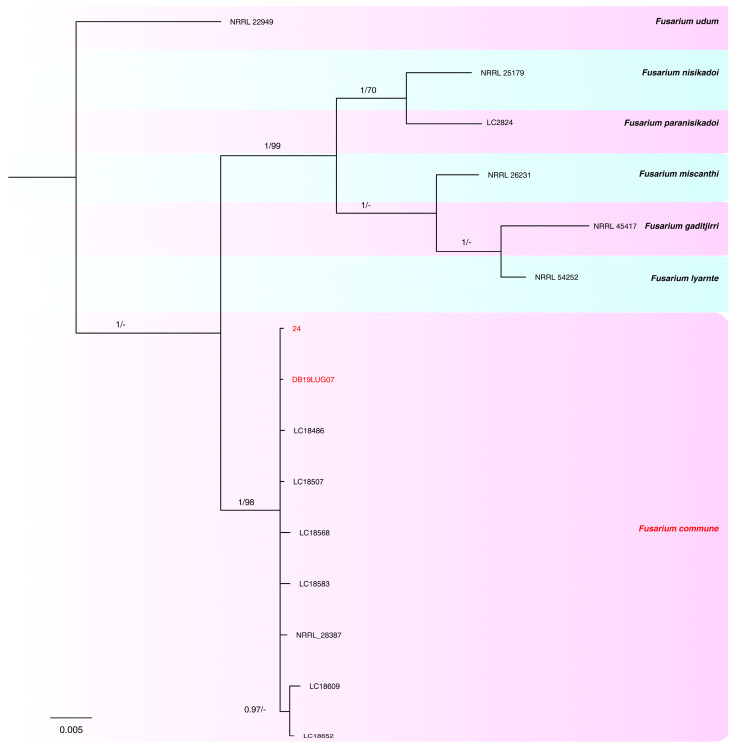
Consensus phylogram of 301 trees resulting from a Bayesian analysis of the combined *tef1-α*, *rpb2*, *calm* and *tub2* sequences of *Fusarium* spp. belonging to FNSC. Bayesian posterior probability values and bootstrap support values are indicated at the nodes. The isolates collected and species found in this study are indicated in red. The tree was rooted to *Fusarium udum* (NRRL22949).

**Figure 5 plants-12-03857-f005:**
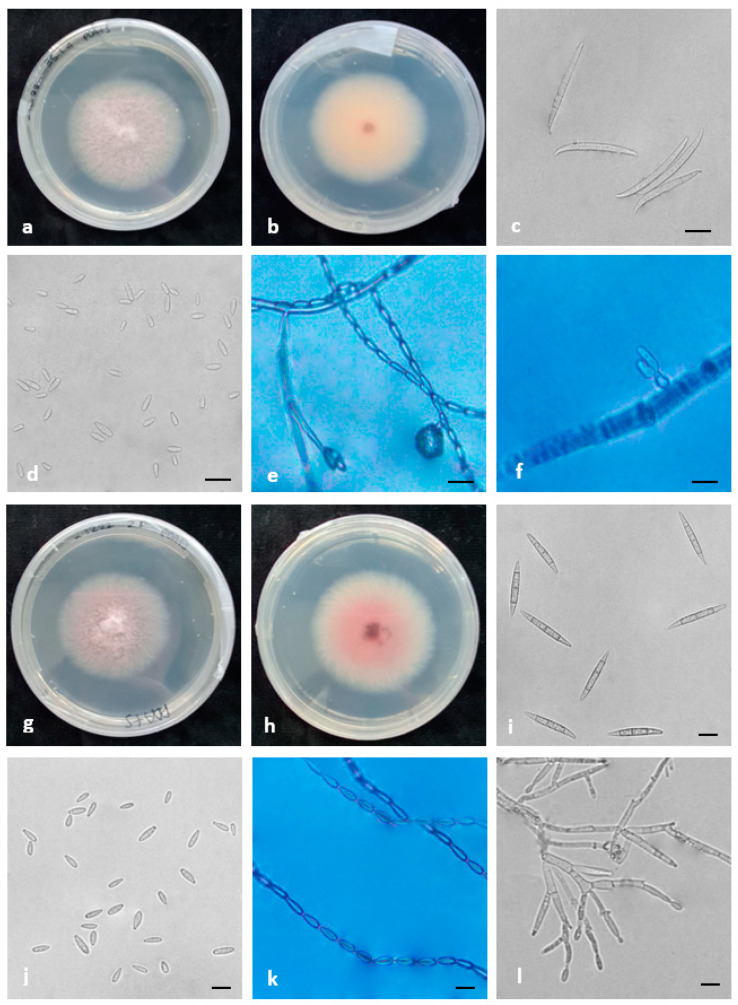
Morphological features of the species belonging to FFSC identified in this study. (**a**–**f**) *F. verticillioides* and (**g**–**l**) *F. annulatum*. (**a**,**b**,**g**,**h**) Colonies on PDA above and below; (**c**–**e**,**i**–**k**) conidia; (**f**–**l**) conidiogenous cells. Scale bars = 10 μm.

**Figure 6 plants-12-03857-f006:**
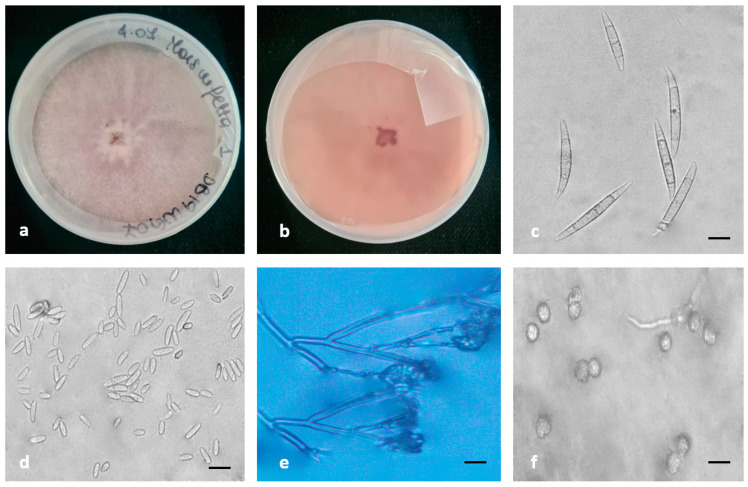
Morphological features of *F. commune*, the species belonging to FNSC identified in this study. (**a**,**b**) Colonies on PDA above and below; (**c**,**d**) conidia; (**e**) conidiogenous cells; (**f**) chlamydospores. Scale bars = 10 μm.

**Figure 7 plants-12-03857-f007:**
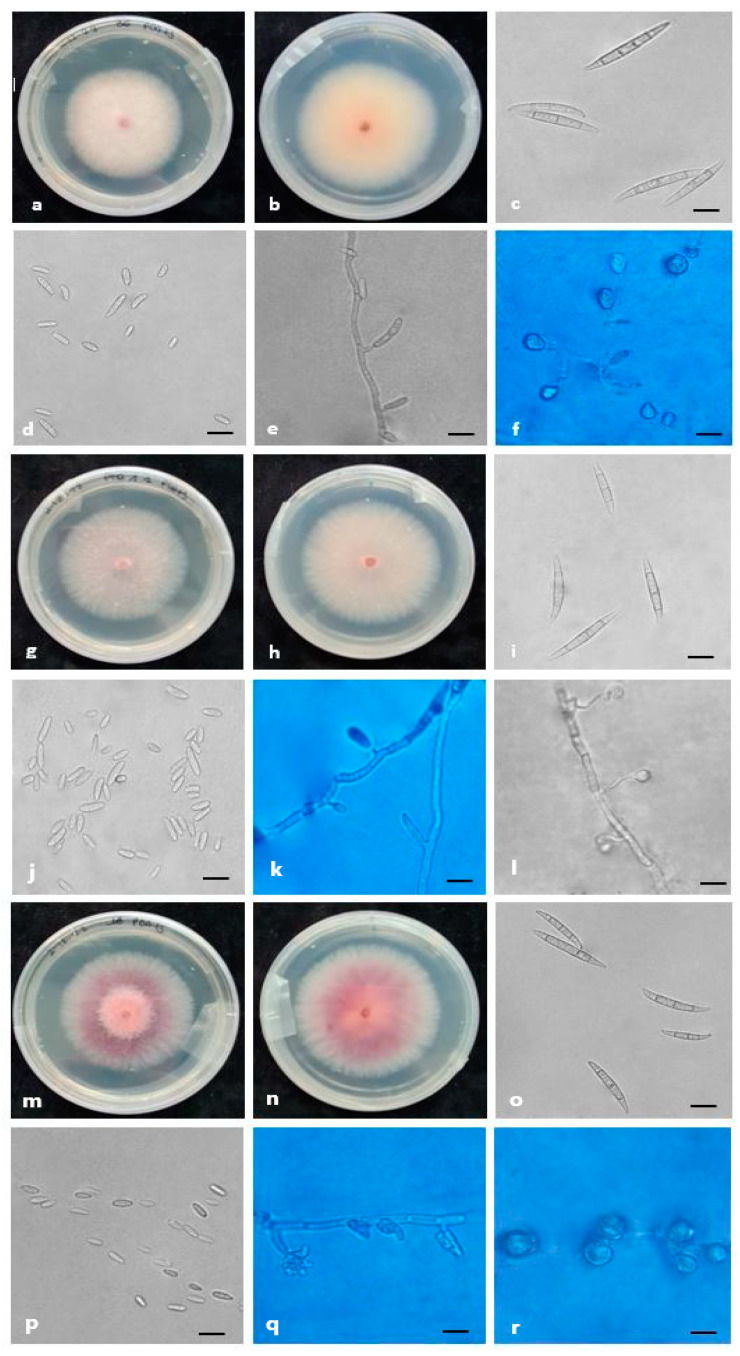
Morphological features of the species belonging to FOSC identified in this study. (**a**–**f**) *F. cugenangense,* (**g**–**l**) *F. nirenbergiae* and (**m**–**r**) *F. oxysporum sensu lato*. (**a**,**b**,**g**,**h**,**m**,**n**) Colonies on PDA above and below; (**c**,**d**,**i**,**j**,**o**,**p**) conidia; (**e**,**k**,**q**) conidiogenous cells; (**f**,**l**,**r**) chlamydospores. Scale bars = 10 μm.

**Table 1 plants-12-03857-t001:** *Fusarium* spp. isolates used in this study (isolate code, origin of the sample, hybrid, FAO class, symptomatic portion used for isolation and year of isolation).

Isolate Code	Origin	Hybrid	FAO Class	Symptomatic Portion	Year of Isolation
DB19LUG07	San Zenone degli Ezzelini (VI)—Italy	Unknown	Unknown	Root	2019
DB19LUG16	San Zenone degli Ezzelini (VI)—Italy	Unknown	Unknown	Root	2019
DB19LUG20	San Zenone degli Ezzelini (VI)—Italy	Unknown	Unknown	Root	2019
DB19LUG25	San Zenone degli Ezzelini (VI)—Italy	Unknown	Unknown	Root	2019
2.1	Livorno Ferraris (VC)—Italy	P1547	600–130 days	Root	2019
2.2	Livorno Ferraris (VC)—Italy	P1547	600–130 days	Root	2019
8.1	Cigliano (VC)—Italy	-	-	Root	2019
8.2	Cigliano (VC)—Italy	-	-	Root	2019
9	USA	PR32B10	600–132 days	Seed	2019
10.1	France	P0423	400–116 days	Seed	2019
10.2	France	P0423	400–116 days	Seed	2019
11	Italy	unknown	unknown	Seed	2019
12	Italy	SY ANTEX	600–130 days	Seed	2019
18	Turkey	DKC6752	600–128 days	Seed	2019
19	Romania	DKC5830	500–x days	Seed	2019
21	Crescentino (VC)—Italy	P1547	600–130 days	Stem	2019
23	Crescentino (VC)—Italy	P1547	600–130 days	Root	2019
24	Crescentino (VC)—Italy	P1916	600–130 days	Root	2019
26	Crescentino (VC)—Italy	P1916	600–130 days	Stem	2019
28	Crescentino (VC)—Italy	P1916	600–130 days	Root	2019
29	Cigliano (VC)—Italy	P1517W	600–128 days	Root	2019
30	Cigliano (VC)—Italy	P1517W	600–128 days	Root	2019
31	Cigliano (VC)—Italy	P1517W	600–128 days	Stem	2019
32	Cigliano (VC)—Italy	P1517W	600–128 days	Stem	2019
35.1.4	Cigliano (VC)—Italy	P1517W	600–128 days	Root	2019
36	Cigliano (VC)—Italy	P1517W	600–128 days	Stem	2019
40	Cigliano (VC)—Italy	P1517W	600–128 days	Root	2019
41	Cigliano (VC)—Italy	P1547	600–130 days	Root	2019
44	Cigliano (VC)—Italy	P1547	600–130 days	Root	2019
50	Cigliano (VC)—Italy	P1547	600–130 days	Root	2019
51	Cigliano (VC)—Italy	Unknown	Unknown	Stem	2019
55.2.1	Cigliano (VC)—Italy	Unknown	Unknown	Crown	2019
56.1.2	Cigliano (VC)—Italy	Unknown	Unknown	Root	2019
56.2.2	Cigliano (VC)—Italy	Unknown	Unknown	Root	2019
56.2.3	Cigliano (VC)—Italy	Unknown	Unknown	Root	2019
56.2.4	Cigliano (VC)—Italy	Unknown	Unknown	Root	2019
56.2.5	Cigliano (VC)—Italy	Unknown	Unknown	Root	2019
57.2.1	Cigliano (VC)—Italy	Unknown	Unknown	Root	2019
1.RI (Pta 1.1)	San Zenone degli Ezzelini (VI)—Italy	Unknown	Unknown	Crown	2020
1.RI (Pta 1.2)	San Zenone degli Ezzelini (VI)—Italy	Unknown	Unknown	Crown	2020
1.RII (Pta 3.2)	San Zenone degli Ezzelini (VI)—Italy	Unknown	Unknown	Crown	2020

**Table 2 plants-12-03857-t002:** Results of pathogenicity test performed on the 41 *Fusarium* isolates isolated, at 14 days. The severity index of root and crown rot is reported as number of recorded plants. The disease index (0–100) of each isolate was calculated. Letters refers to Duncan post-hoc test (*p* < 0.05%) performed after one way ANOVA.

ID Sample	Severity Index of Root and Crown Rot (Number of Plants)					Disease Index	
	1	2	3	4	5	(DI) 0–100	
DB19LUG07	0	3	3	0	0	50.0	abcde
DB19LUG16	0	6	0	0	0	40.0	cdefg
DB19LUG20	4	2	0	0	0	13.3	gh
DB19LUG25	3	3	0	0	0	20.0	gh
2.1	0	0	0	4	2	86.7	a
2.2	0	0	0	3	3	90.0	a
8.1	6	0	0	0	0	0.0	h
8.2	3	3	0	0	0	20.0	fgh
9	0	0	0	6	0	80.0	ab
10.1	0	0	2	0	4	86.7	a
10.2	0	0	1	2	3	86.7	a
11	3	3	0	0	0	20.0	efgh
12	0	3	0	0	3	70.0	abc
18	2	3	0	0	1	36.7	efgh
19	6	0	0	0	0	0.0	h
21	2	4	0	0	0	26.7	efgh
23	2	4	0	0	0	26.7	efgh
24	0	3	0	0	3	70.0	abc
26	0	4	2	0	0	46.7	bcdef
28	3	3	0	0	0	20.0	efgh
29	0	6	0	0	0	40.0	cdefg
30	3	3	0	0	0	20.0	efgh
31	2	4	0	0	0	26.7	efgh
32	4	2	0	0	0	13.3	gh
35.1.4	0	1	1	2	2	76.7	abc
36	3	3	0	0	0	20.0	efgh
40	0	4	2	0	0	46.7	bcdef
41	6	0	0	0	0	0.0	h
44	6	0	0	0	0	0.0	h
50	6	0	0	0	0	0.0	h
51	2	2	2	0	0	33.3	defgh
55.2.1	0	1	1	2	2	76.7	abc
56.1.2	0	0	0	4	2	86.7	a
56.2.2	0	0	2	2	2	80.0	ab
56.2.3	0	0	0	3	3	90.0	a
56.2.4	0	0	2	2	2	80.0	ab
56.2.5	0	0	2	4	0	73.3	abc
57.2.1	0	0	0	4	2	86.7	a
1.RI (Pta 1.1)	2	2	0	0	2	46.7	cdefg
1.RI (Pta 1.2)	0	2	2	0	2	66.7	abcd
1.RII (Pta 3.2)	3	3	0	0	0	20.0	efgh
Healthy control	6	0	0	0	0	0.0	h

## Data Availability

Data are contained within the article and Appendix A.

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
