# Peer review of "Diversity and Pathogenicity of Fusarium Species Associated with Stalk and Crown Rot in Maize in Northern Italy"

_plants, 2023, doi:10.3390/plants12223857_

Round 1
Reviewer 1 Report
Comments and Suggestions for Authors
This manuscript describes diversity and pathogenicity of Fusarium species associated with stalk and crown rot on maize in Northern Italy. The data presented in this manuscript is interesting and the methods used are reasonable. The manuscript, however, has several critical issues. I would recommend this manuscript for publication in Plants, provided the authors reasonably address the following points.
1. Only 36 pathogenicity isolates were obtained, so how many samples were collected in each maize field? And how many roots, stems or crowns were collected in each sample?
2. Only Fusarium spp. was isolated? What about other genus fungi?
3. A total of 2210 characters were embodied in the phylogenetic analysis of the concatenated 4-gene dataset for FFSC isolates. However, the sum of the number of parsimony-informative character (563), variable and parsimony-uninformative character (604) and constant character (1025) was 2977. Why? This is also for FOSC isolates and FNSC isolates.
4. In line 442-444, five primer pairs were listed but only 4 loci were used for phylogenetic analyses.
5. Furthermore, alignments and trees should be deposited in TreeBase and accession provided.
Comments on the Quality of English LanguageMinor editing of English language required.
Reviewer 2 Report
Comments and Suggestions for Authors
Review of manuscript: Diversity and pathogenicity of Fusarium species associated with stalk and crown rot on maize in Northern Italy, by Sanna et all.
I believe the research presented by the authors is valuable and well-written. However, I noticed that the presentation of the results lacks an important practical aspect related to the pathogenicity of Fusarium isolates and their species membership. Although the authors consider pathogenicity to be significant, the coherence was missing from the results description and the study summary. I found a brief mention of this topic in lines 355-360. In my opinion, a statistical analysis is necessary to confirm which species are more important as pathogens of maize. The results of this analysis should be presented in the manuscript as a table or graph, which will help in writing the conclusions summarizing the research done by the authors. This analysis should also be included in the abstract of the paper.
I believe the manuscript can be published with the following changes:
1. In line 23-24, please specify the pathogenicity of the Fusarium isolates.
2. In Table 2, please perform a statistical analysis taking into account the species affiliation of the Fusarium isolates. Present the results as a table or graph, and include the result in the conclusion of your work.
3. In Fig. 2 and 3, please specify the meaning of the red color on the diagram and enter the codes of the isolates from your research into the diagrams.
4. In lines 380-401, please specify whether only Fusarium fungi were grown from the plant material, if not, describe how the initial selection was carried out.
5. In lines 381-382, please provide the geographical coordinates.
6. In L391, please specify whether the seed material was purchased, whether it was a single variety, and whether the quality parameters of the seed material tested are known.
7. In lines 423-425, please cite the relevant literature.
8. In L428, please change "Data analysis" to "Statistical analysis".
9. Please transfer Table 3 to supplementary material.
Reviewer 3 Report
Comments and Suggestions for Authors
Please see attached

Double check grammar and spelling
Round 2
Reviewer 2 Report
Comments and Suggestions for Authors
The manuscript has been corrected. All issues have been clarified. I accept the publication of the paper. I congratulate the authors on their interesting research.